# Glycidamide Promotes the Growth and Migratory Ability of Prostate Cancer Cells by Changing the Protein Expression of Cell Cycle Regulators and Epithelial-to-Mesenchymal Transition (EMT)-Associated Proteins with Prognostic Relevance

**DOI:** 10.3390/ijms20092199

**Published:** 2019-05-04

**Authors:** Titus Ime Ekanem, Chi-Chen Huang, Ming-Heng Wu, Ding-Yen Lin, Wen-Fu T. Lai, Kuen-Haur Lee

**Affiliations:** 1Ph.D. Program for Cancer Molecular Biology and Drug Discovery, College of Medical Science and Technology, Taipei Medical University and Academia Sinica, Taipei 11529, Taiwan; titusekanem@yahoo.com; 2Department of Hematology, University of Uyo, Uyo 520221, Nigeria; 3Graduate Institute of Neural Regenerative Medicine, College of Medical Science and Technology, Taipei Medical University, Taipei 11031, Taiwan; hcc0609@tmu.edu.tw; 4Graduate Institute of Translational Medicine, Taipei Medical University, Taipei 11031, Taiwan; mhwu1015@tmu.edu.tw; 5Department of Biotechnology and Bioindustry Sciences, College of Bioscience and Biotechnology, National Cheng Kung University, Tainan 003107, Taiwan; lindy@mail.ncku.edu.tw; 6Graduate Institute of Cancer Biology and Drug Discovery, College of Medical Science and Technology, Taipei Medical University, Taipei 11031, Taiwan; 7McLean Imaging Center, McLean Hospital, Harvard Medical School, Belmont, MA 02478, USA; laitw@tmu.edu.tw; 8Department of Research, Taipei Medical University/Shuang-Ho Hospital, New Taipei City 23561, Taiwan; 9Department of Dentistry, Taipei Medical University/Shuang-Ho Hospital, New Taipei City 23561, Taiwan; 10Ph.D. Program for Cancer Molecular Biology and Drug Discovery, College of Medical Science and Technology, Taipei Medical University, Taipei 11031, Taiwan

**Keywords:** glycidamide, prostate cancer, cell cycle regulator, EMT, gene signature, prognosis

## Abstract

Acrylamide (AA) and glycidamide (GA) can be produced in carbohydrate-rich food when heated at a high temperature, which can induce a malignant transformation. It has been demonstrated that GA is more mutagenic than AA. It has been shown that the proliferation rate of some cancer cells are increased by treatment with GA; however, the exact genes that are induced by GA in most cancer cells are not clear. In the present study, we demonstrated that GA promotes the growth of prostate cancer cells through induced protein expression of the cell cycle regulator. In addition, we also found that GA promoted the migratory ability of prostate cancer cells through induced epithelial-to-mesenchymal transition (EMT)-associated protein expression. In order to understand the potential prognostic relevance of GA-mediated regulators of the cell cycle and EMT, we present a three-gene signature to evaluate the prognosis of prostate cancer patients. Further investigations suggested that the three-gene signature (CDK4, TWIST1 and SNAI2) predicted the chances of survival better than any of the three genes alone for the first time. In conclusion, we suggested that the three-gene signature model can act as marker of GA exposure. Hence, this multi-gene panel may serve as a promising outcome predictor and potential therapeutic target in prostate cancer patients.

## 1. Introduction 

Certain substances in diets, such as acrylamide (AA) and glycidamide (GA), have been reported to induce malignant transformations [1,2]. AA can be produced in carbohydrate-rich food when heated at high temperatures (> 120 ℃) [3,4]. In addition, commonly consumed food and beverages that are reported to contain AA include processed cereals, potato chips and coffee [5]. It has been reported that AA is a carcinogen in animal experiments [6]. In cell models, the proliferation rates of some cancer cell lines, such as endometrial and ovarian cancer cells have been shown to increase upon treatment with AA and GA [1]. The genes which have growth-promoting potential, such as the oncogenes, have been demonstrated to be induced upon treatment with AA or GA in normal human primary cells [1,7]. However, although an increase the proliferation rate caused by treatment with AA and GA has been proven in some cancer cells, the exact genes that were induced by AA or GA in most of the cancer cells were not clarified. In addition, in intestinal tumourigenesis mouse models, it was shown that GA increased the number of small intestinal tumors by a larger amount relative to AA [8]. Thus, a better understanding of the carcinogenesis of GA may lead to a potential therapeutic strategy for cancer patients who have suffered from GA-induced mutagenesis.

In humans, dietary intake of AA has been associated with a high risk of various of cancers, including brain, lung, breast, colorectal, ovarian, endometrial cancer, and prostate cancers [9,10,11,12,13,14]. However, the conclusion of these studies shows that only colorectal, ovarian, and endometrial cancer risk may be elevated in association with dietary AA. In addition, several recent epidemiological studies, focused on dietary exposure to AA and cancer incidence, have reported conflicting or inconclusive results, as reviewed in [15]. It has been argued that the possibility to detect an increased cancer risk from dietary intake of AA by epidemiology is very small as there is exposure to everyone, assumed to contribute to small increments in the individual risks, without large variations in the population [16]. In addition, it is difficult to determine the individual intakes of AA, because AA is present in many foods with large variations in the concentration [17]. Moreover, the latest research has demonstrated GA mutational signature is found in a full one-third of approximately 1600 tumor genomes corresponding to 19 human tumor types from 14 organs. [18]. This study provide new insights into the thus-far the association of GA with human carcinogenesis. The genotoxic effects of GA have been described in all of the cancers mentioned above, except in prostate cancer [10,18,19,20,21,22]. In prostate cancer, it was shown that regular consumption of certain deep-fried foods is associated with an increased prostate cancer risk [23]. AA is widespread in these foods, with an estimated 38% of calories consumed in the US coming from foods that contain AA [24]. It was also shown that AA can undergo metabolic oxidation via cytochrome P450 2E1 (CYP2E1) to the genotoxic epoxide GA [25], and is considered as the putative genotoxic agent in AA exposure [26]. However, the carcinogenic role of GA in prostate cancer tumorigenesis and the kind of genes that are altered by GA in prostate cancer cells are still unclear.

Prostate cancer is a disease associated with advanced age in males. There are two subtypes, namely, the slow-growing and the aggressive cancers. Currently the five-year survival rate for the slow-growing subtype is about 100%; however, thereafter patients die from the metastatic disease [27]. There is always a lack of response to the instituted therapy for the aggressive subtype. Thus, the high sensitivity of tumor markers for the detection of the aggressive type of prostate cancer is the most essential determinant of survival. It has been demonstrated that cyclin D1 and CDK4 play central roles in the regulation of proliferation of prostate cancer [28,29]. In addition, the epithelial–to -mesenchymal transition (EMT) is a process by which epithelial cells lose their cell polarity and cell-cell adhesion, and gain migratory and invasive properties. Moreover, EMT-related transcription factors (EMT-TFs), such as TWIST1 (Twist), SNAI1 (Snail), and SNAI2 (Slug) were all required for migration/metastasis and the alteration of these factors was associated with advanced prostate cancer [30,31]. As a key epithelial marker responsible for adherens junction, CDH1 (E-cadherin) enables the cells to maintain epithelial phenotypes which can be carried out by EMT-TFs [32]. The loss of CDH1 (E-cadherin) expression is associated with metastatic progression of prostate cancer [33]. Thus, these EMT markers have the potential to be significant prognostic factors in predicting prostate cancer survival.

In the present study, we demonstrated that GA-promoted the growth ability of prostate cancer cells through induced protein expression of the cell cycle regulator. In addition, we also found that GA-promoted the migratory ability of prostate cancer cells through induced epithelial-to-mesenchymal transition (EMT)-associated protein expression. Moreover, we further investigated the GA-mediated change of the above proteins, which could be used as a biomarker for prostate cancer outcomes.

## 2. Results 

### 2.1. Effects of GA on the Doubling-Time, Cell Viability, and Cell Mobility of Prostate Cancer Cells

To understand the effects of GA on the growth characteristics of prostate cancer cells, the doubling-times of LNCap, DU145 and PC3 cells treated with or without GA were determined. Cell numbers at different treatment time-points (0, 24, 48, 72 and 96 h) were determined by using a hemacytometer and the trypan blue dye-exclusion method. As shown in Table 1, GA reduced the doubling-times of LNCap, DU145 and PC3 cells from 37.6 h, 35 h, and 35 h, respectively, to 15.2 h, 14.2 h, and 13.2 h. Next, the viability of these three prostate cancer cell lines was further determined. The chemical structure of GA is shown in Figure 1A. As shown in Figure 1B, compared to untreated cells, the strong pro-proliferative effects were observed after incubation with GA (1 μM) for 72 h in three prostate cancer cell lines. In addition, it has been demonstrated that changes of gene expression with up to 2-fold inductions or repressions could be observed following a 1 μM GA treatment, thus approaching the concentration range that is relevant for the assessment of human health risks [7]. Therefore, 1 μM of GA was chosen to determine the migratory abilities of the GA-treated prostate cancer cells using a migration transwell assay. As shown in Figure 1C–D, the migratory abilities significantly increased by about by 3-fold, 2-fold, and 2.5-fold in GA-treated LNCap, DU145 and PC3, respectively. Taken together, these results showed that GA promotes cell growth and migration and contributes to malignant phenotypes of prostate cancer cells.

### 2.2. Alteration of Regulators of Cell Cycle and the Epithelial–to–Mesenchymal Transition (EMT) in GA-Treated Prostate Cancer Cells

To further investigate whether GA promotes cell growth and migration through alteration of the expression of cell cycle regulators and EMT-TFs, Western blotting for cell cycle regulators and EMT-TFs was examined. As shown in Figure 2, the induction of cyclin D1 and CDK4 protein expression was observed in GA-treated with LNCap cells (see Figure 2A), DU145 cells (see Figure 2B), and PC3 cells (see Figure 2C). In addition, GA-treated prostate cancer cells also caused the induction of EMT-TFs (twist, snail, and slug) and vimentin expression in conjunction with a concomitant decrease in the expression of the E-cadherin that was observed in three prostate cancer cell lines (see Figure 2). Together, these findings suggest that GA can enhance the induction of cell growth and migration of prostate cancer cells by inducting the expression of cell cycle regulators and EMT-TFs.

### 2.3. GA-Mediated mRNA Expression of Regulators of the Cell Cycle and EMT in Prostate Cancer Tissues 

Further, to understand whether GA-mediated activation of cell cycle regulators and EMT-TFs were upregulated in prostate cancer tissues, we investigated expressions of these genes in a public prostate cancer tissue dataset. These genes’ mRNA expression profiles were obtained using existing complementary DNA microarray datasets under accession no.GSE21032 [34] deposited in the Gene Expression Omnibus (GEO). In a Taylor microarray dataset of the GEO website, there were 29 normal prostate glands, 131 primary prostate cancer tissues, and 19 metastatic tissues. As shown in Figure 3, we observed that mRNA expressions of CDK4 (see Figure 3B), TWIST1 (see Figure 3C), and SNAI1 (Snail) (see Figure 3D) were significantly upregulated in metastatic prostate cancer tissues compared to the primary tumor group, whereas mRNA expressions of the CCND1 (Cyclin D1) (see Figure 3A), and CDH1 (E-cadherin) (see Figure 3F), show no statistically significant difference between primary prostate cancer tissues and metastatic prostate cancer tissues (*p* = 0.277 for CCND1; *p* = 0.440 for CDH1). However, mRNA expression of SNAI2 (Slug) (see Figure 3E), showed significant downregulation in metastatic prostate cancer tissues compared to the primary tumor group.

### 2.4. Prognostic Relevance of GA-Mediated mRNA Expression of Regulators of the Cell Cycle and EMT in Prostate Cancer Tissues

We next explored the prognostic relevance of GA-mediated cell cycle regulators and EMT-TFs in prostate cancer using SurvExpress survival analysis [35]. The patients from the GSE21032 dataset [34] (*n*=140) were classified into predicted low- and high-risk groups according to the Prognostic Index (PI; see Appendix A).The results demonstrated that low expression of CCND1, CDK4 and TWIST1 were correlated with low risk (see Appendix A), good prognosis and longer survival time (see Figure 4A–C), while high expression of SNAI1, SNAI2, and CDH1 indicated low risk (see Appendix A), good prognosis and longer survival time (see Figure 4D–F). Survival differences between predicted low- and high-risk groups were evaluated using Kaplan–Meier survival curves. *P*<0.05 was considered to be statistically significant. Collectively, these results indicated that there was low expression of CCND1, CDK4 and TWIST1; high expression of SNAI1, SNAI2, and CDH1 was correlated with good prognosis of prostate cancer patients.

### 2.5. CombinationThree-Gene Signature Predicted Survival in Prostate CancerPatients

The expression alteration of the abovementioned genes was identified to be associated with the prognosis of prostate cancer patients. However, the efficacy of a single gene index was limited; multi-gene-combination prediction can improve the sensitivity of clinical outcomes of cancer patients [36]. Thus, combinations of multi-gene models of prostate cancer patients were analyzed using Kaplan–Meier survival analysis. CDK4, TWIST1, and SNAI2 three-genes were selected based on the significant expression profiles of these genes (see Figure 3), and prognostic relevance of these genes for prostate cancer patients (see Figure 4). Specifically, as shown in Appendix A, significant differences in genes selected by a combination of any two-gene models in clinical outcomes were exhibited according to the Kaplan–Meier survival analysis; in particular, the most significant model was the CDK4, TWIST1, and SNAI2-three-gene signature combination. In our three-gene signature, the PI of the 140 patients ranged from 3.707 to 8.047, with an optimal cut-off value of 7.286, which is described in Section 4. A PI of less than 7.286 was divided into the low-risk group (n = 125), while a PI higher than 7.286 was considered to be a high-risk group (n = 15). The analysis demonstrated that a low risk was correlated with low expression of CDK4 and TWIST1 and high expression of SNAI2, while a high risk was correlated with high expression of CDK4 and TWIST1 and low expression of SNAI2 (see Figure 5A). In addition, we detected the gene expression level of CDK4, TWIST1, and SNAI2 in the high-risk and low-risk groups. Our results show that the gene expressions of CDK4 and TWIST1 were higher in the high-risk group than that in the low-risk group, while the gene expressions of SNAI2 was lower in the high-risk group than that in the low- risk group. All were found to have significant difference in the three-gene signature (*p* = 4.75 × 10^−6^ for CDK4, *p* = 4.73 × 10^−5^ for TWIST1, and *p* = 1.52× 10^−11^ for SNAI2; see Figure 5B). Moreover, Kaplan–Meier survival curves showed that patients with a predicted low risk (*n* = 125) had a significantly longer survival time than those with high risk (*n* = 15) (*p* = 6.876 × 10^−8^; see Figure 5C). Taken together, our results suggested that the most significant model of the three-gene signature was related to survival and was a predictor of the prognosis of prostate cancer. This may help to provide significant clinical implications for the prognosis prediction of prostate cancer in patients suffering from GA-induced mutagenesis.

## 3. Discussion 

In this study, we identified, for the first time, that GA-treated prostate cancer cells exhibited aggressive features. There are several lines of evidence that support this phenomenon. First, GA reduced the doubling-times and had strong pro-proliferative effects on prostate cancer cells. Second, GA promoted the migratory abilities of prostate cancer cells, which may contribute to malignant phenotypes of prostate cancer cells. Third, GA markedly increased the protein expression of two pro-proliferative cell cycle regulators, three EMT-TFs and vimentin, and decreased protein expression of E-cadherin in three prostate cancer cells. This is significant as these proteins are relevant to aggressive forms of prostate cancer cells. Fourth, GA-mediated the expression of one pro-proliferative cell cycle regulator (CDK4) and two EMT-TFs (TWIST1 and SNAI2) of prostate cancer cells can act as a marker of GA exposure. Collectively, this study is the first to report that protein expression of the CDK4, twist, and slug is induced by GA in prostate cancer cells, and their aberrant expression together accurately predicted prostate cancer patient’s outcomes.

TWIST1 and SNAI2 are two key emerging regulators of malignant progression and EMT [37]. Our finding shows a significant association of TWIST1 mRNA upregulation with metastatic prostate tumor tissues compared to normal and primary prostate tissues. In prostate cancer, increased expression of TWIST1 plays an important role in the development of prostate cancer [38] which is consistent with our result. In addition, our results show that significant association of SNAI2 mRNA downregulation with metastatic prostate tumor tissues compared to normal and primary prostate tissues. SNAI2 gene expression is often downregulated due to methylation of the promoter; the expression of SNAI2 is restored or elevated at the invasion front of high-grade prostate cancer and lymph node metastases [39]. Moreover, Esposito et al. also showed that SNAI2 downregulation observed in most prostate cancer epithelia cells is linked to hypermethylation of the SNAI2 gene promoter [39]. Furthermore, it has been demonstrated that knockdown of SNAI2 can cause downregulation of CCND1 [39]. This result can explain the downregulation of CCND1 in primary prostate cancer groups compared to normal prostate groups (see Figure 3A). 

High–throughput technologies are capable of extensive analysis of gene and protein expression profiles on a larger scale with higher sensitivity as compared to the conventional techniques [40]. Recent studies revealed that some prognostic models based on cancer-related genes were constructed, and the predictive performances of these models were validated in different cancer types [36]. In our study, we analyzed the association of CDK4, TWIST1, and SNAI2 single gene expression with the prognosis of prostate cancer patients in the GSE21032 dataset taken from the SurvExpress database. The data demonstrated that high expression of CDK4 and TWIST1 were significantly correlated with high risk and poor prognosis; while low expression of SNAI2 indicated high risk and poor prognosis (*p*<0.05). However, the efficacy of a single index was limited. Therefore, multi-gene biomarkers may be better suited to capturing the complex effects of heterogeneous diseases, such as cancer, on mRNA abundance levels. Thus, we identified the most significant model of three-gene signature (CDK4, TWIST1, and SNAI2) that was able to predict the survival of prostate cancer patients for the first time. Overall, the exhibition of single-gene and multi-gene biomarkers across many different diseases would assist in the development of a clinically useful outcome prediction from gene expression data

In conclusion, using a multiplexed, molecularly driven approach, we suggest that a three-gene signature model can act as a marker of GA exposure. In addition, this multi-gene panel may serve as promising outcome predictors and potential therapeutic targets in GA-induced prostate cancer patients.

## 4. Materials and Method

### 4.1. Chemicals, Reagents, and Antibodies

Glycidamide, methanol, and crystal violet were obtained from Sigma (St. Louis, MO, USA). A quantity of 100 mg glycidamide was dissolved in 100 µl methanol and 1.050 mL deionized water and stored at −20°C until it was used. Rabbit antibodies against Cyclin D1, CDK4, twist, snail, slug, vimentin, and E-cadherin were obtained from Cell Signaling (Beverly, MA, USA). Mouse monoclonal antibody against β-actin was purchased from MP Biomedicals (Irvine, CA, USA).

### 4.2. Cell Cultures

PC3 and LNCaP cell lines (ATCC, Manassas, VA, USA) were cultured in RPMI-1640 (Thermo Fisher Scientific; Waltham, MA, USA), supplemented with 10% fetal bovine serum (FBS) (Thermo Fisher Scientific; Waltham, MA, USA) and antibiotics (Thermo Fisher Scientific; Waltham, MA, USA). The DU145 (ATCC) cell line was cultured in the minimum essential amount of Eagle’s medium (Thermo Fisher Scientific; Waltham, MA, USA) supplemented with 10% FBS, 2 mM L-glutamine (Thermo Fisher Scientific; Waltham, MA, USA), and antibiotics. Cultures were maintained in a 5% CO_2_ humidified atmosphere at 37 °C.

### 4.3. Doubling-Time Determination.

The experimental cells were seeded at a density of 1.8 × 10^5^. Cell numbers at different treatment time-points (0, 24, 48, 72 and 96 h) were determined by using a hemacytometer and the trypan blue dye-exclusion method. Doubling time was calculated using the formula, doubling time = t × log2/(logN_t_ − logN_0_), where N_t_ represents number of cells at time t, N_0_ represents initial number of cells, as described previously [41].

### 4.4. Cell Viability Assay

Cell viability was determined using the crystal violet staining method, as described previously [42]. In brief, the cells were plated in 96-well plates at 3000 cells/mL and treated with/without glycidamide at the indicated concentrations. Viable cells were stained with 0.5% crystal violet in 30% ethanol for 10 min at room temperature. Subsequently, the plates were washed four times with tap water. After drying, the cells were lysed with a 0.1 M sodium citrate solution (Sigma, St. Louis, MO, USA), and the dye uptake was measured at 550 nm using a 96-well plate reader. Cell viability was calculated by comparing the relative dye intensities of the treated and untreated samples.

### 4.5. Western Blotting

Cell lines were placed in a lysis buffer at 4 °C for 1 h. Protein samples were electrophoresed using 8–15% SDS-polyacrylamide gel electrophoresis, as described in [43]. 

### 4.6. In Vitro Migration Assay

Assays were performed using Falcon™ cell culture inserts (8-μm pore size) in a 24-well plate (BD Biosciences, San Jose, CA, USA) according to the manufacturer’s instructions. In the migration assay, the cells (10^4^ cells/well) in 0.5 mL of serum-free medium were seeded onto the upper chamber membranes that received different treatment. These membranes were previously inserted into the 24-well plates containing 10% FBS-supplemented medium. After 24 h, the cells were fixed with 100% methanol and stained with 5% Giemsa stain (Merck, Darmstadt, Germany). Non-migrated cells that remained in the upper chambers were removed by wiping the top of the inserted membranes using a damp cotton swab, leaving only those cells that migrated to the underside of the membranes. All experiments were performed in triplicate and photographed under a phase-contrast microscope (200×).

### 4.7. Oncomine Database Analysis

Analysis of the change in gene expression in prostate cancer tissues was performed by using the online cancer microarray database Oncomine, (www.oncomine.org, Compendia biosciences, Ann Arbor, MI, USA). The threshold search criteria used in the study were a *p*-value < 0.0001, a fold change > 2, and a gene rank in the top 5%.

### 4.8. SurvExpress Database Analysis

In our analysis, SurvExpress was used to provide survival analysis and risk assessment. SurvExpress (http://bioinformatica.mty.itesm.mx/SurvExpress), which is a comprehensive gene expression database, can provide risk assessment and survival analysis in cancer datasets using a biomarker gene list as an input. The samples of each dataset were split into two risk groups, for which each group was determined according to the ordered prognostic index (PI; high value denoting a high risk) [35]. SurvExpress can perform risk grouping through an optimization algorithm using the ordered PI. For example, a log-rank test was accomplished using the arranged PI values for the two risk groups. Then, the algorithm selected the dividing point, where *P* was at the minimum value. The PI was computed using the expression levels and values estimated from the Cox fitting algorithm [44].

### 4.9. Statistical Analysis

*P*-values and fold-changes for differential expression analysis of genes generated from the Oncomine database were calculated using a one-sided Student’s *t*-test. Statistical analyses of the cell viability assay and migration assay were performed by using an unpaired Student’s *t*-test in the Excel software. *P* values < 0.05 were considered significant.

## Figures and Tables

**Figure 1 ijms-20-02199-f001:**
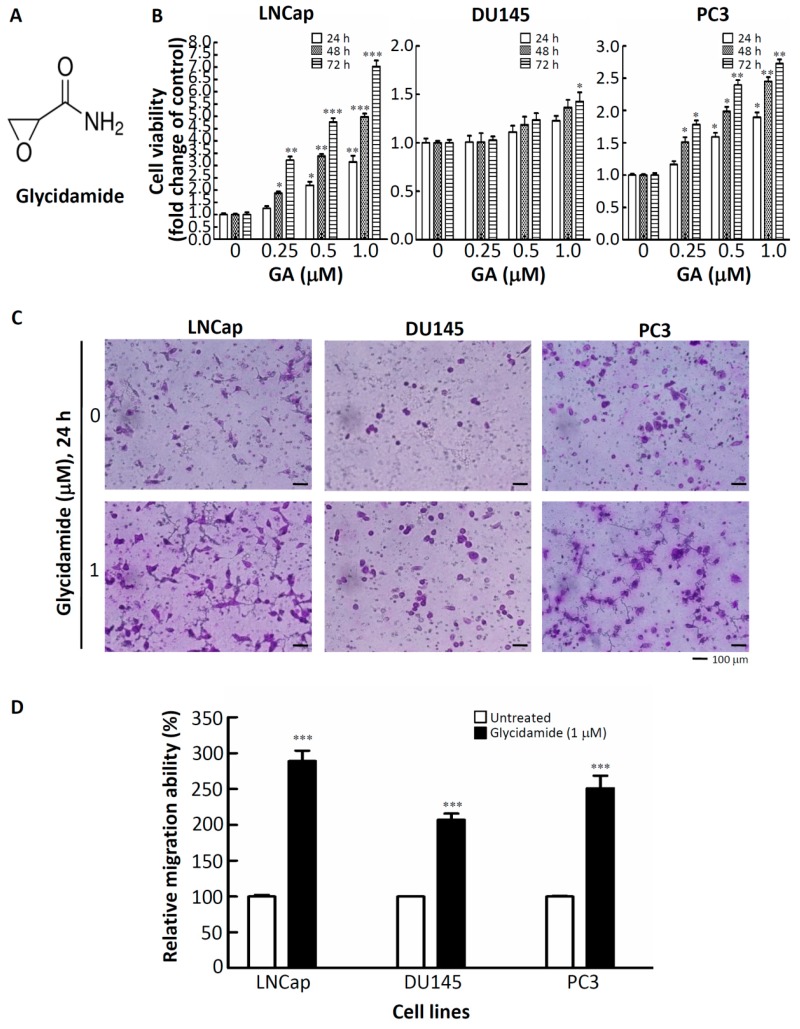
Glycidamide (GA) promotes cell growth and migration and contributes to malignant phenotypes of prostate cancer cells. (**A**) Chemical structure of glycidamide (GA). (**B**) The effects of GA on cell viability were determined in a panel of prostate cancer cell lines. Cells were treated with GA at the indicated concentrations in a10% fetal bovine serum (FBS)-supplemented medium for 24, 48, and 72 h, and cell viability was assessed using a crystal violet staining method. Bars, SD (*n* = 6). (**C**,**D**) The effects of GA on the migratory activity of a panel of prostate cancer cell lines after 24 h of treatment. * *p* < 0.05, ** *p* < 0.01, *** *p* < 0.001. Values are presented as the mean ± SD of three independent experiments. SD: standard deviation.

**Figure 2 ijms-20-02199-f002:**
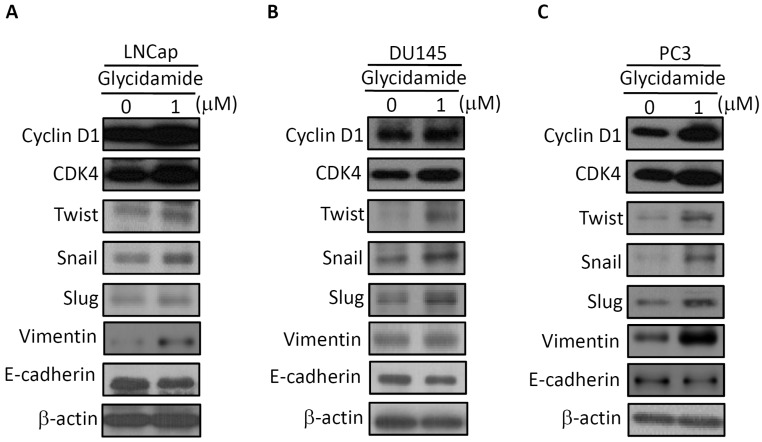
The effect of GA on protein expression of regulators of the cell cycle and epithelial-to-mesenchymal transition (EMT). LNCap (**A**), DU145 (**B**), and PC3 (**C**) cell lines were treated with GA at the indicated concentrations for 48 h. Cell extracts were analyzed by Western blotting with the antibodies of cyclin D1, CDK4, twist, snail, slug, vimentin, and E-cadherin, respectively. All experiments were performed in three independent experiments.

**Figure 3 ijms-20-02199-f003:**
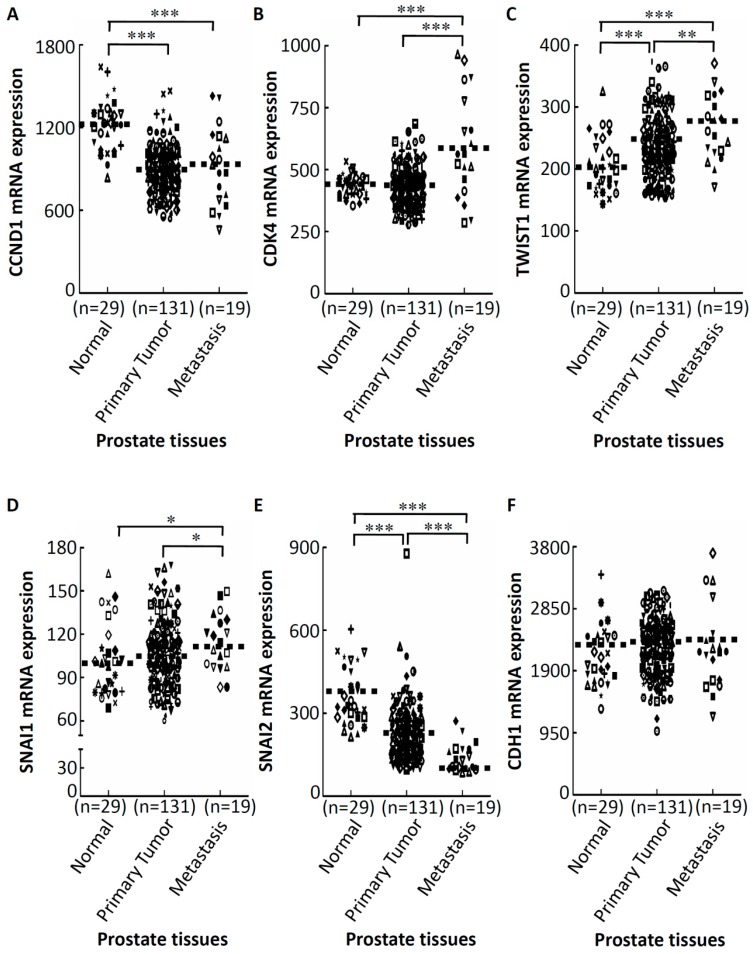
Aberrant expressions of GA-modulated cell cycle-related genes and EMT-related genes expression of prostate cancer patients. Relative expression levels of CCND1 (**A**), CDK4 (**B**), TWIST1 (**C**), SNAI1 (**D**), SNAI2 (**E**), and CDH1 (**F**) in different clinical stages of prostate cancer tissues analyzed using the public Gene Expression Omnibus (GEO) database (GSE21032). * *p* < 0.05, ** *p* < 0.01, *** *p* < 0.001.

**Figure 4 ijms-20-02199-f004:**
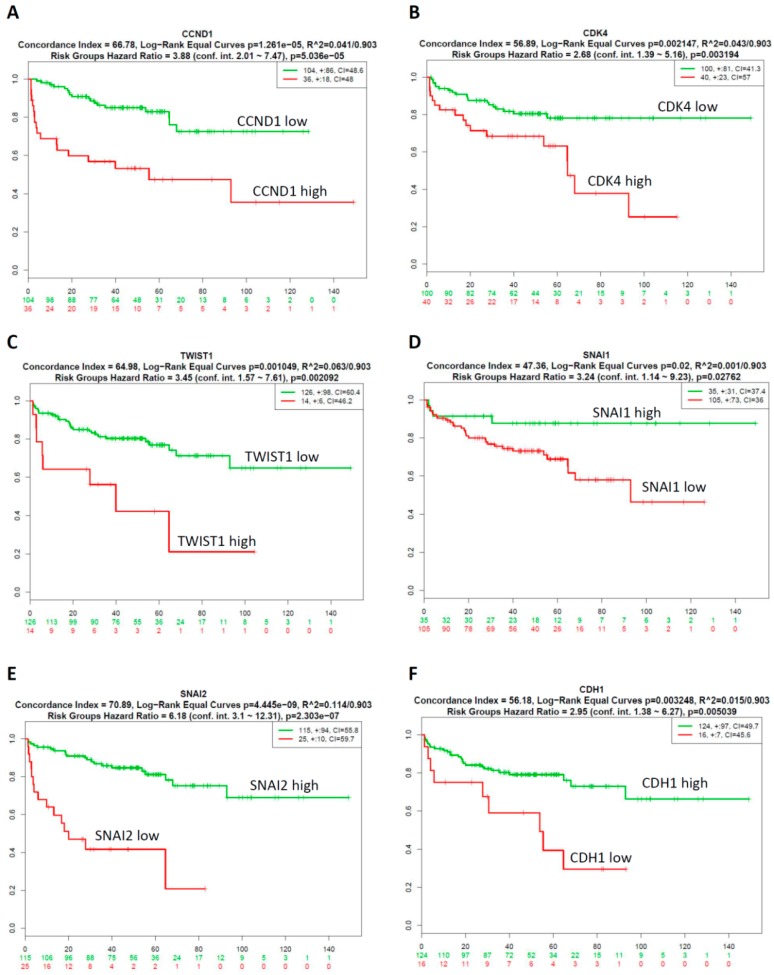
Correlation between GA-modulated cell-cycle-related genes or EMT-related genes expression and the survival rates in prostate cancer patients. Low expression of CCND1 (**A**), CDK4 (**B**), and TWIST1 (**C**) were correlated with longer survival time of prostate cancer patients. High expression of SNAI1 (**D**), SNAI2 (**E**), and CDH1 (**F**) were correlated with longer survival time of prostate cancer patients. Green and red lines indicate low- and high-risk groups, respectively. *P* < 0.05 was considered to be statistically significant.

**Figure 5 ijms-20-02199-f005:**
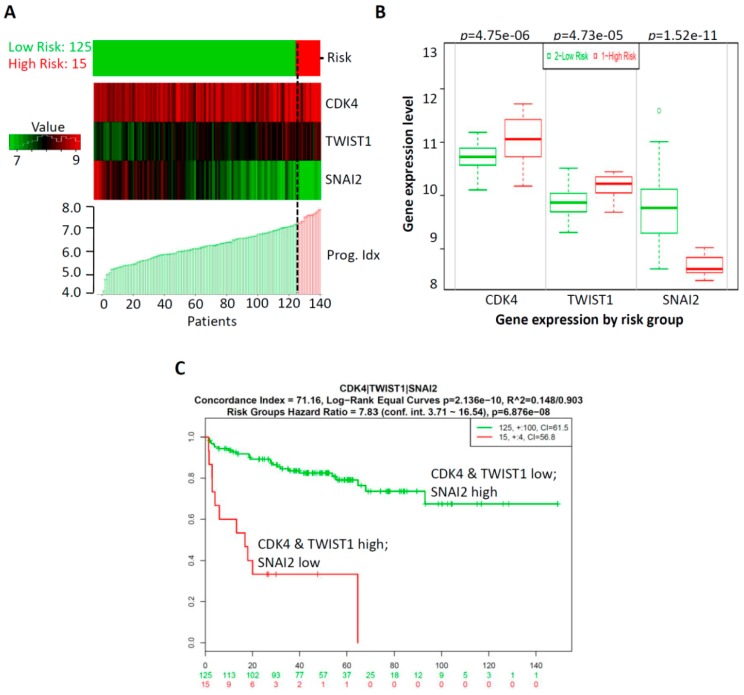
The three-gene signature predicted survival better than the individual genes alone in prostate cancer patients. (**A**) The SurvExpress database was used to analyze the association of the three-gene signature with the predicted risk. (**B**) The gene expression level of CDK4, TWIST1, and SNAI2 were detected in the high-risk and low-risk groups. (**C**) Kaplan–Meier survival curves showed that patients with a predicted low risk (n = 125) had a significantly longer survival time than those with a high risk (n = 15). *P* < 0.05 was considered to be statistically significant.

**Table 1 ijms-20-02199-t001:** The inoculated and harvested densities and doubling times of glycidamide-treated prostate cancer cells.

Cell Type	Inoculated Density	Harvested Density	Doubling Time (h)
LNCap-N	1.8 × 10^5^	6.8 × 10^5^	37.6
LNCap-G	1.8 × 10^5^	4.8 × 10^6^	15.2
DU145-N	1.8 × 10^5^	7.5 × 10^5^	35.0
DU145-G	1.8 × 10^5^	6.1 × 10^6^	14.2
PC3-N	1.8 × 10^5^	7.5 × 10^5^	35.0
PC3-G	1.8 × 10^5^	7.8 × 10^6^	13.2

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
