# Peer review of "Glycidamide Promotes the Growth and Migratory Ability of Prostate Cancer Cells by Changing the Protein Expression of Cell Cycle Regulators and Epithelial-to-Mesenchymal Transition (EMT)-Associated Proteins with Prognostic Relevance"

_ijms, 2019, doi:10.3390/ijms20092199_

Round 1

Reviewer 1 Report

In what appears to be a technically sound, straightforward set of in vitro experiments, the authors investigated the effects of glycidamide (GA), a carcinogenic metabolite of acrylamide (AA), on a panel of three well-characterized, widely used prostate cancer cell lines.  Specifically, they showed that GA (data shown for only one dose, 1 microM) reduced cell doubling time, increased cell viability (by MTT assay), and enhanced migration (Transwell migration assay) in LNCaP, DU-145 and PC-3 human prostate cancer cells.  This was followed by western blot analysis of regulators of cell cycle and EMT in cells treated with GA.  From this data, they conclude that GA increased the abundance of cyclin D1, CDK4, Twist, Snail and Slug, and decreased expression of E-cadherin, consistent with the functional data on proliferation and migration.

Interestingly, they then accessed the NCBI GEO database to use mRNA microarray data from prostate tumor samples and matched normal tissue samples to examine gene expression of these same cell cycle and EMT regulators in normal prostate tissues, primary prostate tumors, and metastatic prostate cancer tissues.  Consistent with their WB data, they found that CDK4, Twist1 and SNAI1 (Snail) were upregulated in metastatic vs primary tumors.  In contrast, CCND1 (cyclin D1) and SNAI2 (Slug) were downregulated and CDH1 (E-cadherin) showed no change.  Then, they analyzed these changes in gene expression in light of clinical outcomes and determined that a 3-gene signature of CDK1, Twist and SNAI2 predicted survival in these prostate cancer patients. 

Issues to be addressed:

1. Rationale for examining effects of GA in prostate cancer needs to be clarified/strengthened.  

a) As rationale for studying the effects of GA on prostate cancer cells, the authors cite a number of published studies that investigated the association between dietary AA and cancer risk in different malignancies, including prostate cancer.  About these studies, the authors state, “In human, dietary intake of AA has been associated with high risk of various of cancers, including brain, lung, breast, colorectal, ovarian, and prostate cancers [9-14]” (Page 2, Line 15).  However, this statement appears to be inaccurate in that not all of the articles provided in support of this statement are, in fact, supportive.  While these reports do indeed examine the potential role of dietary AA in the risk of the specific malignancies listed (brain, lung, colorectal, ovarian, prostate), as well as endometrial cancer, which is not listed by the authors, these studies conclude that only endometrial, ovarian and colorectal cancer risk may be elevated in association with dietary AA.  In fact, the authors of Ref #13 concluded they "found no evidence that acrylamide intake, within the range of U.S. diets, is associated with increased risk of prostate cancer."  Moreover, two reviews of the literature on dietary AA and cancer concluded that meaningful interpretation of such studies cannot be made until exposure assessment methods have been improved (Nutr Cancer. 2014; 66(5): 774–790) and, more critically, that continued epidemiological investigation of AA and cancer risk is misguided (Eur J Cancer Prev. 2012; 21:375–386).

The authors will need to amend their assessment of the literature in the field and provide a clear rationale for examining the effects of GA in the context of prostate cancer.

2. Rationale for GA dose selection needs to be clarified/strengthened.  

a) Data in Figs. 1 and 2 are shown for a single dose of GA (1 microM).  What is the rationale for selecting this dose and, importantly, how does it relate to human exposures?  Note that Ref #7, which the authors cite to support the ability of GA to regulate expression of genes related to tumor progression, actually concludes that such changes occur at high doses that exceed typical human dietary exposure.

3. The authors’ intent regarding the link between GA exposure and the prognostic 3-gene signature needs clarification. The authors identified a 3-gene signature in metastatic tumor tissue that correlated with patient survival; specifically, upregulation of CDK4 and Twist and downregulation of SNAI2.  These gene targets were selected based on in vitro data showing changes in expression of the corresponding proteins in GA-treated prostate cancer cells.

a) Page 6, Line 16: "to understand whether GA-mediated activation of cell cycle regulators and EMT-TFs were upregulated in prostate cancer tissues, we investigated expressions of these genes in public prostate cancer tissue dataset."

Page 12, Line 14: “GA-mediated induction of one pro-proliferative cell cycle regulator (CDK4) and two EMT-TFs (TWIST1 and SNAI2) of prostate cancer cells were associated with a shorter survival of prostate cancer patients.” 

Page 12, Line 36: "three-gene signature was related to the survival of prognosis in prostate cancer patients with suffered GA."

These statements and similar ones in the manuscript need to be clarified.  It seems that the authors are trying to link human GA exposure to the 3-gene signature in patient samples and thus to patient prognosis.  But, the data do not support the contention that these changes in gene expression can be linked to GA exposure in human subjects.  The fact that this gene signature appears to be associated with patient survival is intriguing, but the data do not indicate that these changes in the human samples are GA-specific or related to GA exposure.   

Other issues:

Language:

1.  The paper is in need of language editing.  Some examples are listed below:

a) Inappropriate expression/usage:

(i) Page 2, Line 36: “GA reduced the doubling-times of LNCaP, DU145 and PC3 cell growth by 15.3 h, 14.2 h, and 13.2 h, respectively, to compare with the untreated-LNCaP, DU145 and PC3 cells by 37.6 h, 35 h, and 35 h.”

The word “by” is used incorrectly.  The doubling times are not reduced by the amounts of 15.3 h, 14.2 h and 13.2 h; the reduced doubling times ARE 15.3 h, etc.  The sentence should say something like, “GA reduced the doubling-times of LNCaP, DU145 and PC3 cells from 37.6 h, 35 h, and 35 h, respectively, to 15.3 h, 14.2 h, and 13.2 h.”

(ii) Page 2, Line 39: “prostate cancer cell lines were further used to determine the viability of GA.”

The viability of the cells was determined, not the viability of GA.

   (iii) Page 12, Line 43: “was able to predict the survival of prostate cancer prognosis”

Predicting the “survival of prostate cancer prognosis” does not make sense.

 b) Spelling:

(i) Page 4, Line 6: “Alternation” should be “Alteration.”

 c) Incomplete sentences:

(i) Page 10, Line 2: “The expression alternation of the above-mentioned genes which we identified to be associated with prognosis of prostate cancer patients.”

No verb is present before the “which” phrase.

 d) Unclear meaning of expression:

(i) Page 12, Line 46: “we suggested that three-gene signature was related to the survival of prognosis in prostate cancer patients with suffered GA.”

Meaning is unclear.  Is the intended meaning that the 3-gene signature was related to the survival of patients exposed to GA?  Or, that the signature is a marker of GA exposure?  Or something else?

Materials and Methods:

1.  4.3 Cell Proliferation Assay:  Please indicate here or in the relevant figure legend the specific time intervals that were tested to determine doubling time, or at least the longest interval tested.

2.  4.4 Cell Viability Assay and 4.5 Western Blotting: Please indicate in the Materials and Methods or in the relevant figure legends how long cells were treated for each of these analyses.

3.  4.6 In Vitro Migration Assay: In this section, it is stated that cells were treated for 12 h before measurement of migration, but in the corresponding figure legend (Fig. 1), it states 24 h.

Other:

1. Page 10, Line 13: An optimal cut-off PI value of 7.286 was used to distinguish low vs high risk groups.  Please describe here or in the Materials and Methods how this cut-off value was determined.

2. Statistical analysis was performed on the migration data, but not on the viability data.  Please provide analysis for the viability data.

3. Please indicate how many times each of the in vitro experiments was performed and whether the data shown represent means of biological replicates or technical replicates.

Author Response

Response in attached PDF file

Reviewer 2 Report

Please add a list of abbreviations used in the text.

There are a lot of grammar errors and an extensively english revision is needed.

The concentration of GY used in the experiments is very high. What is the GY concentration released from food?

 In the introduction section, there is not a description of the EMT role in prostate cancer.  

The methods are described improperly.  For example,  in the  paragraph "cell viability assay" or in "the cell proliferation assay" there is not a description of MTT assay.

In the first set of experiments, authors used three types of prostate cancer cell lines that differ for the expression of androgen or estrogen receptors. Is there a relationship between GY and steroid receptors mechanism of Action?

Please add a reference for the method used to establish the doubling time in table 1.

In Figure 1B is not reported the standard deviation and the time of treatment. I suggest an evaluation of cell viability at different time and with different GY concentrations. 
Lines 8-12 at page 4 should be moved in the introduction section.

An other marker,  for example,  vimentin should be added to western blot analysis in Figure 2. Additionally, other methods to evaluate migration should be performed. Recently it has been published a lot of articles on EMT in prostate cancer that could be useful to the authors.
The results reported in the paragraphs 2.3, 2.4 and 2.5 are not consistent with results reported in the first paragraphs. They analyzed prostate cancer samples and normal prostate gland but there is not a correlation with exposure to GY. Moreover the results can not support the conclusions.

Author Response

Response in attached PDF file

Round 2

Reviewer 2 Report

Authors improved the manuscript according to my suggestions.